# Global Incidence and Mortality Patterns of Pedestrian Road Traffic Injuries by Sociodemographic Index, with Forecasting: Findings from the Global Burden of Diseases, Injuries, and Risk Factors 2017 Study

**DOI:** 10.3390/ijerph17062135

**Published:** 2020-03-23

**Authors:** Moien A. B. Khan, Michal Grivna, Javaid Nauman, Elpidoforos S. Soteriades, Arif Alper Cevik, Muhammad Jawad Hashim, Romona Govender, Salma Rashid Al Azeezi

**Affiliations:** 1Department of Family Medicine, College of Medicine and Health Sciences, United Arab Emirates University, Al Ain 15551, UAE; jhashim@uaeu.ac.ae (M.J.H.); govenderr@uaeu.ac.ae (R.G.); 201400972@uaeu.ac.ae (S.R.A.A.); 2Primary Care, NHS North West London, London TW3 3EB, UK; 3Institute of Public Health, College of Medicine and Health Sciences, United Arab Emirates University, Al Ain 15551, UAE; m.grivna@uaeu.ac.ae (M.G.); javaid.nauman@uaeu.ac.ae (J.N.); esoteria@uaeu.ac.ae (E.S.S.); 4Department of Circulation and Medical Imaging, Faculty of Medicine and Health Sciences, Norwegian University of Science and Technology, 7491 Trondheim, Norway; 5Harvard School of Public Health, Department of Environmental Health, Environmental and Occupational Medicine and Epidemiology (EOME), Boston, MA 02115, USA; 6Internal Medicine, Emergency Medicine Section, College of Medicine and Health Sciences, United Arab Emirates University, Al Ain 15551, UAE; aacevik@uaeu.ac.ae

**Keywords:** road traffic injuries, pedestrians, public safety, global burden, economic loss, health policy, healthcare system, health indicators

## Abstract

(1) Background: Pedestrian injuries (PIs) represent a significant proportion of road traffic injuries. Our aim was to investigate the incidence and mortality of PIs in different age groups and sociodemographic index (SDI) categories between 1990 and 2017. (2) Method: Estimates of age-standardized incidence and mortality along with trends of PIs by SDI levels were obtained from the Global Burden of Disease from 1990 to 2017. We also forecasted the trends across all the SDI categories until 2040 using the Statistical Package for the Social Sciences (SPSS Statistics for Windows, version 23.0, Chicago, IL, USA) time series expert modeler. (3) Results: Globally, the incidence of PIs increased by 3.31% (−9.94 to 16.56) in 2017 compared to 1990. Men have higher incidence of PIs than women. Forecasted incidence was 132.02 (127.37 to 136.66) per 100,000 population in 2020, 101.52 (65.99 to 137.05) in 2030, and reduced further to 71.02 (10.62 to 152.65) by 2040. Globally across all SDI categories, there was a decreasing trend in mortality due to PIs with the global estimated percentage reduction of 37.12% (−45.19 to −29.04). (4) Conclusions: The results show that PIs are still a burden for all SDI categories despite some variation. Although incidence and mortality are expected to decrease globally, some SDI categories and specific vulnerable age groups may require particular attention. Further studies addressing incidence and mortality patterns in vulnerable SDI categories are needed.

## 1. Introduction

Every day, about 1800 pedestrians never return home due to road traffic injuries and fatalities [1]. An estimated 12 million pedestrian road traffic injuries (PIs) occur on an annual basis [1]. PIs are a major public health problem across all ages and sexes, accounting for nearly one quarter of all road traffic-related injuries [2]. Among all road injuries, PIs carry the highest risk of a person being severely injured, leading to significant morbidity, disability, and death [3,4]. Such a burden inflicts pain and suffering on injured pedestrians and their families and has an economic impact costing approximately 0.5% of the total world Gross National Product and 130 billion US dollars globally [5].

This surge has been due to rapid urbanization leading to an increased amount of motor vehicles as well as changes in the environment and lifestyle due to global economic development [6]. Risk factors of PIs are well described and can vary in different age groups and settings. For example, children’s cognitive development influences their ability to make safe decisions [7]. Adolescents maybe actively seeking out risk [8]. Moreover, longevity and aging lead to increased dependency on personal automobiles for work- and non-work-related activities, which may increase associated risks [9].

Critical reasons for the increase in PIs and a slower progress to reduce such occurrences could be the variation of socioeconomic factors and disparities between different age groups and regions globally. The ambitious United Nations goal to reduce injuries and deaths due to road traffic collisions by 50% in five years from 2015 to 2020 has not been achieved [10,11]. The new 2030 Agenda, referred to as the Sustainable Development Goals (SDGs), included road safety as an important part of global development and a crucial responsibility for governments, corporations, and civil society [11]. 

Studying global and social variation trends of PIs is essential to mitigate key risk factors. Additionally, such a detailed review will aid the adoption of evidence-based practices and solutions to reduce PIs and related deaths. Furthermore, quantifying a socioeconomic relationship with PIs would help to identify age groups, gender, countries, and regions where increased risk is documented, thereby allowing systematic planning to implement appropriate measures to reduce such risks. Although there has been a detailed report on road traffic injuries, very few studies have quantified PIs by SDI globally [12]. This study aims to investigate the incidence and mortality patterns of PIs in different SDI categories between 1990 and 2017. It also provides forecasts up to 2040. 

## 2. Methods

### 2.1. Data Source and Data Extraction

The latest Global Burden of Disease, Injuries, and Risk factors study 2017 (GBD 2017) is a detailed, multinational, epidemiological study with enhanced methodology providing regional and global estimates. We extracted the data on PIs from the latest GBD 2017 dataset, which is available in the public domain. Ethics approval or permission is not required to use the GBD 2017 data. Age-standardized rates were used to allow comparisons of populations amongst different age groups. International Classification of Diseases (ICD) codes ICD-9 and ICD-10 were used by the GBD 2017 study to extract data for PIs [13,14]. In GBD 2017, the data were obtained from several sources, including public health research studies, vital registrations, verbal autopsies, police records, trauma registries, country-based surveys, health insurance records, emergency departments, hospital out-patient records, and hospital claim records. The PI estimation process in the GBD 2017 study used epidemiological meta-regression tools to produce incidence and mortality by age, gender, year, and location. The detailed methodology was explained in the methods description for the GBD 2017 [15].

The data sources were later systematically quantified using various injury modeling strategies and statistical methods by age, gender, year, country, and cause. A detailed outline concerning the methods of collecting and calculating data for the GBD 2017 study is provided elsewhere [16,17].

### 2.2. Sociodemographic Index

The sociodemographic index (SDI) is a composite metric of overall development, which has a strong positive relationship with health outcomes [18]. The measure quantifies and differentiates the countries and regions based on the spectrum of development. It encompasses three different aspects of development and is a geometric mean of income per capita, educational achievement of the population, and the fertility rate [19]. SDI is interpreted on a scale of 0 to 1, with a value of 1 indicating the highest development and value 0 indicating the lowest development. The SDI divides the world regions into five quintiles based on their SDI values from 0 to 1, i.e., low SDI (LSDI), low–middle SDI (LMSDI), middle SDI (MSDI), high–middle SDI (HMSDI), and high SDI (HSDI). A detailed methodology with regard to the calculation of SDI cutoff points is reported elsewhere [17].

### 2.3. Outcome Metrics

The worldwide burden due to PIs was estimated by analyzing several outcomes, including age-standardized incidence and mortality with 95% uncertainty intervals (UI). These measures were also analyzed by age groups and segregated according to the world SDI set by the GBD 2017 study. A complete set of PI incidence and mortality, with trends and forecasting, was computed for the years 1990, 2000, 2010, and 2017. The percentage change (degrees of change) was analyzed at different points at various times between the years 1990 and 2017. Furthermore, the percentage difference between the two values was calculated to show changes in age- and SDI-specific incidence and mortality between the years 1990 and 2017.

### 2.4. Uncertainty Intervals

Uncertainty estimations could be due to incomplete information, potential biases in information, or heterogeneity among information sources, data generation process, and model uncertainty [20]. Such measurement errors affecting the data input are expressed as the uncertainty interval (UI). The UI represents a range of values that reflects the certainty of an estimate. Entire measures have been reported with a 95% UI.

### 2.5. Statistical Analysis

The Statistical Package for the Social Sciences (SPSS Statistics for Windows, version 23.0, Chicago, IL, USA) was used for statistical analysis. We estimated the percentage difference of incidence and mortality due to PIs for every SDI category with the available GBD 2017 estimates between two time points. We calculated the 95% UIs with the 2.5th and 97.5th percentiles range for the change based on the cause-specific model estimation for each GBD 2017 SDI, age group, gender, and time points between 1990 and 2017 [21]. The SPSS time series modeler with the expert modeler option was used to predict the future trends of PI incidence and mortality. None of the observed values during forecasting were marked as outliers.

## 3. Results

### 3.1. Incidence and Trends

In 2017, the global estimated incidence of PIs per 100,000 population was 140.92 (115.81 to 168.53) (Table 1A). This translates into 11.5 million pedestrian injuries occurring worldwide. The highest incidence was seen among the HMSDI category 195.52 (160.55 to 234.45), while the lowest PIs occurred in the HSDI category 110.05 (88.33 to 135.33). The estimates of incidence rates and trends for PIs by SDI in 1990, 2000, 2010, and 2017 are shown in Table 1A.

Globally, PI incidence increased by 3.31% (−9.94 to 16.56) in 2017 compared to 1990. HSDI countries showed the most considerable decrease in incidence, 21.39% (−35.03 to −7.74), between 1990 and 2017, while MSDI countries showed a significant increase, 26.62% (8.87 to 44.36), in the same period (Table 2A). Within the last seven years (2010–2017), the HMSDI showed the greatest reduction of PIs, 12.38% (−20.64 to −4.11). In the same period, the global reduction was 6.58% (−14.25 to 1.09). In contrast, the HSDI category showed the lowest percentage decrease of 1.79% (−9.75 to 6.17) between 2010 and 2017 (Table 2A). 

### 3.2. Incidence and Trends by Age and Sex

Between 1990 and 2017 globally, men had a higher incidence of PIs, 169.47 (139.27 to 202.93), compared to women 112.37 (92.36 to 134.14). HMSDI countries showed the highest rate of PIs in men compared to other SDI categories. Incidences and trends of PIs by SDI are shown in Figure 1.

Globally, all trend lines were ascending from 1 to 20 years of age while there was a steady occurrence between 20 and 75 years. The second increase in trend lines was seen after 75 years of age. The trend line pattern was similar between 1990 and 2017 (Figure 2). The highest incidence was seen in the age group of 80 to 84 years, 247.56 (184.29 to 325.79). Above the age of 75 years, an average of 210 incidents were recorded per 100,000 population. Of those under 65 years of age, the 25 to 29 age group recorded the highest incidence of 162.94 per 100,000. There were incidence and trend variations between each SDI region and age group. Trends for the age-specific prevalence of PIs by SDI categories are presented in Figure 2.

### 3.3. Forecasted Incidence Trends

We expect a decreasing trend between 2020 and 2040, globally and across all regions (Figure 3). 

We forecasted an incidence rate of 132.02 (127.37 to 136.66) in 2020, compared to 101.52 (65.99 to 137.05) in 2030 and 71.02 (−10.62 to 152.65) in 2040 per 100,000 population. Surprisingly, the HMSDI category showed the highest forecasted incidence rate of 172.8 (62.52 to 183.08), compared to the lowest forecasted in the HSDI category of 107.72 (103.9 to 111.54) for the year 2020. The highest expected incidence was seen in 2030 is the MSDI category of 127.56 (65.96 to 189.16), and the lowest expected incidence rate was seen in the LSDI of 78.16 (35.38 to 120.94). In 2040, the MSDI is still higher than the global average of 115.25 (−33.2 to 263.71), and, interestingly, the lowest expected incidence in 2040 is to be seen in the HMSDI of 15.11 (−165.59 to 195.8).

### 3.4. Mortality and Trends

In 2017, the estimated mortality was 6.25 (5.77 to 6.94) per 100,000 population, which is translated into nearly half a million deaths globally due to PIs. The LSDI category recorded the highest mortality rate of 8.29 (7.24 to 9.64), while HSDI regions showed the lowest mortality rate of 1.69 (1.64 to 1.76) in 2017. The mortality rates and trends for PIs by SDI in 1990, 2000, 2010, and 2017 are shown in Table 1B and Table 2B.

Globally across all SDIs, there was a decreasing trend in mortality due to PIs with a global estimated reduction of 37.12% (−45.19 to −29.04) between 1990 and 2017. However, the highest reduction was seen in the HSDI category of 62.19% (−70.48 to −53.89), as compared to the reduction in the LSDI category of 27.78% (−34.61 to −20.94) (Table 2). In the last seven years, the highest reduction in mortality was observed in the HMSDI category of 25.12% (−30.19 to −20.04). The lowest reduction was in the LSDI category of 14% (−19.41 to −8.58). 

### 3.5. Mortality Trends by Age and Sex

In Figure 4, we show the trends for age-specific mortality across SDI categories between 1990 and 2017. Globally, there is an increasing trend starting from the age of 10 to 14 years. Though mortality has reduced globally across all the regions, it is essential to note that the patterns in age trends have not changed in the last three decades.

In Figure 5, we depict the mortality trends by sex per 100,000 population. Globally, across all SDI categories, men had a higher mortality rate than women over time. Globally, the death rate for men in 2017 was 8.90 (8.21 to 9.95) compared to a 3.62 (3.35 to 3.95) death rate for women, which equates to approximately 150% more deaths in men. Furthermore, in 2017, the smallest difference in mortality rate between sexes was observed in the HSDI category, where it was notably twice as high in men at a rate of 2.32 (2.25 to 2.42), compared to 1.07 (1.03 to 1.11) in women.

### 3.6. Forecasted Trends of Mortality

In Figure 6, we show the forecasted age-standardized mortality rates globally by SDI categories. 

The prediction across SDI regions indicates a generalized decrease in trends in the age-standardized mortality rates. In HSDI categories, we observe a plateau in mortality as opposed to a comparatively sharper decline in other SDIs. Though the trend in declining mortality rates is observed in the LSDI category, the rate is much slower when compared with the other SDI categories. Globally the predicted death rate in 2020 is 5.53 (5.1 to 5.96) per 100,000 population. Globally, we are forecasting approximately 280,000 deaths in 2020, which are still preventable. In 2030, the expected global mortality rate is 3.3 (0.31 to 6.28) and 1.06 (−5.69 to 7.81) in 2040. The predicted mortality trend in 2030 with the SDI categories in LSDI is 5.29 (1.44 to 9.13), 3.58 (−0.54 to 7.7) in the MSDI, 4.64 (1.92 to 7.37) in the LMSDI, 1.7 (−8.77 to 12.17) in the HMSDI, and 1.67 (0.69 to 2.66) in the HSDI per 100,000 population. In 2040 however, across all regions, the death rate is close to one per 100,000 population with the exception of LMSDI, which appears to be twice the global average, and in the LSDI category, where the death rate is still three per 100,000 population.

## 4. Discussion

The current study showed that incidence, trends, and mortality rates of PIs varied between different SDI categories. HMSDI countries showed the highest incidence, while LSDI countries showed the highest mortality in 2017. Our forecast revealed that incidence and mortality rates will continue to decrease until the predicted year 2040. Age-standardized and sex-specific incidence and mortality rates showed similar patterns in each SDI category. The 20 to 75 age group is highly prone to have a pedestrian injury in each SDI category but those mostly affected are children and young adults, with a further peak in mortality being noted from age 70 to 80. Furthermore, males are highly prone to have a pedestrian injury in each SDI category.

Global data on pedestrian injuries are often poorly reported by most countries; thus, the actual statistics for the most vulnerable road users may be higher than reported. In 2013, at least one-fifth of pedestrians died in road traffic collisions [2,22]. The Global Status Report on Road Safety published in 2018 by the World Health Organization (WHO) states that 23% of pedestrians worldwide died because of road traffic collisions. Overall pedestrian mortality from 2013 to 2018 was almost constant, which may be attributed to progress in road safety [1]. There is considerable geographical heterogeneity with respect to pedestrian mortality, with the Americas, Europe, West Pacific, and Southeast Asia having lower mortality rates at 22%, 27%, 22%, and 14%, respectively, and the Eastern Mediterranean region and Africa having the highest pedestrian fatality rates at 34% and 40%, respectively [1]. It must be noted that these variations in pedestrian mortality rates across regions are related to variations in numbers of road users, with pedestrian deaths in low-income countries being disproportionally higher than in high-income countries when adjusted to population and motor vehicle numbers [1]. The assumption that pedestrian mortality may be under-reported emphasizes the need for more accurate data collection from certain regions.

Globally, there were 1,243,068 deaths out of 54,192,330 road injuries in 2017 [10], which equates to a 2.3% mortality rate. Pedestrians are vulnerable road users and carry the highest risk of being severely injured [2,3,4,22,23]. According to the WHO report (2013), 99% of motor vehicles in the world were located in high (47%) and middle–high income countries (52%), while only 1% were in low-income countries. However, the HMSDI category has the highest incidence of PIs compared to the HSDI category. This might be because of the paucity of developments affecting pedestrian injury or the lack of implementation of preventive measures. Furthermore, although low-income countries have a limited number of vehicles and the LSDI category showed the lowest incidence in 2017, its mortality was the highest among all categories. These results should be evaluated differently than HSDI and HMSDI results. For example, we could speculate that LSDI country mortality figures could be due to a lack of injury prevention efforts, absence of pedestrian sidewalks and crossings, or medical facilities [24].

There are various factors affecting incidence and mortality of pedestrian injuries, including vehicle design, speed control, road infrastructure, and traffic law enforcement [6]. Although incidence and mortality vary between countries [25], literature supports that the aforementioned factors are similar worldwide regardless of a country’s SDI category. Grouping by SDI allows the measuring of injury within the context of similar sociodemographic development status. Such group stratification may help develop appropriate strategies to reduce pedestrian injuries and implement policies for prevention.

Although, incidence of PIs increased by 3.31% in 2017 compared to 1990, the decline in incidence rates after 2010 is noteworthy. Starting from 1990, the global PIs incidence rates were similar until year 2005, increased until year 2011, and then again showed a decline until 2017. Our forecasting analyses until 2040 showed a decline in global PI incidence rates. Similar to the trends we showed in this study, pedestrian injuries in various contexts amongst children and the elderly show high rates of severe injuries [26,27,28,29]. Children in preschool and elementary schools show increased incidence. Moreover, children who are socially disadvantaged or have low socioeconomic status are more prone to pedestrian injury compared to those in schools with a high socioeconomic status [30,31], which correlates with the fact that people living in affluent regions are less likely to be injured compared to those living in deprived regions [32]. Likewise, males are dominant in children’s PIs [33]. Children do not need high-speed traffic incidents to be injured or die, with children under 3 years being at high mortality risk even with low-speed incidents, as their somatosensory system is underdeveloped. Interestingly, children under 12 years are not able to use visual and vestibular information as effectively as adults [34,35]. Similarly, the elderly are also more susceptible to PIs and subsequent mortality due to a reduced reflex time, decreased sensory-motor coordination, and age-related auditory and visual impairment, resulting in a slow walk time when crossing streets [29,36,37,38,39]. In addition, we are seeing a rise in geriatric injury as the population ages [28]. Sadeghi-Bazargani reported that the elderly are almost seven times more likely to die due to PIs [40]. In fact, the elderly show the highest mortality trend in almost every SDI category in our study. These results are similar to Demetriades’ study [41], which demonstrates that injury severity increases with age. They found injury severity scores of 15 or higher in 36.8% of the elderly compared to the 11.2% of children [41].

There are various mechanisms and causes of PIs that are mentioned in the literature. Motorcycle collisions with pedestrians are highlighted as being responsible for the majority of deaths [20,34,39,40,42]. However, some studies highlight other factors such as buses, trucks, freeways, weather conditions, time of day, road density, lack of appropriate signaling, zebra crossings, crossing the highway, walking on the pavement, bus stops, and speed [27,29,38,43,44]. 

Although controlling and eliminating all risk factors is not realistic, decreasing the incidence and mortality associated with PIs is possible by modifying behavioral, social, and environmental risk factors [31]. Reports from HSDI countries such as Germany highlight that modifications of car front design reduce pedestrian injury and severity [45]. Not every country has similar developmental stages as HSDI countries, therefore implementing similar modifications to their car industry or importing newly designed vehicles may not be a solution. However, some other simple measures can be taken into consideration including education. McLaughlin reported that increasing awareness and knowledge about PIs among elementary school children with interactive educational sessions reduced PI occurrence by 60% [46]. Some experts highlighted that even as little as one and a half hours of virtual pedestrian environment training reduced the pedestrian error rate in children [47]. Interestingly, educational programs mainly consider children as a target population. However, given in our results we saw that incidence and mortality increased with age, this implies that educational activities should be continued for all age categories and repeated on a periodic basis [48]. The effects of alcohol and other drugs on injuries are undeniable [49]. Increasing alcohol and drug screening among high-risk groups is essential and may also decrease PIs [50]. 

In addition, the implementation of speed limits and road traffic design, along with calming measures is also very effective [29,44,51,52]. A decrease of as little as 10 km per hour in the speed limit has resulted in a favorable decrease of PIs in Japan [44]. Similar findings have been reported from the United Kingdom of a 15% reduction in mean speed leading to reduced injury rates by 21% and fatalities by 75% [53]. A report from Russia highlighted that changing the structure of roads with signalization decreases the PIs rate [54]. 

In our study we predicted that, though the incidence PIs and mortality due to them are decreasing, they are showing a slow reduction over time. The outlined United Nations Sustainable Development Goals targeting a reduction in global injuries and deaths by 50% appear to be overambitious, based on our study, unless robust national and worldwide strategies are introduced according to the needs of each SDI region and country. Otherwise there is a strong possibility that the countries within the LSDI regions will experience higher rates of mortality due to failures to adopt such global plans [55]. 

## 5. Limitations

To our knowledge, this study provides the first overview of global patterns and trends of incidence and mortality of PIs by different SDI regions. However, our results should be reviewed with consideration of the following limitations. Firstly, this study was undertaken using the secondary data from the GBD 2017, which could possibly have inconsistent coding, non-coding, or under-reporting of PIs, leading to overestimation or underestimation of PIs. Such coding errors are especially prevalent in under-resourced SDI regions such as the LSDI and LMSDI countries. However, with the GBD methodology, there is a clear framework to take care of such sampling and non-sampling errors [14]. Moreover, we predicted the future incidence and mortality trends of PIs based on the results obtained from 1990 to 2017. Further studies are required with primary data obtained from nationwide observational studies or trauma registries to verify our findings. Accordingly, trends in the pattern of incidence and mortality may change if national policies, transportation, and socioeconomic parameters related to PIs are modified. 

### Future Work

Many parts of the world are undergoing economic and social development together with the development of road and transportation systems, including public transportation, which could have a major impact on the mortality and morbidity related to PIs. Moreover, national policies coupled with public health campaigns related to road and traffic safety are warranted to better educate both motorists and pedestrians. Future studies related to PIs should include time variations of these factors in their estimates. The advent of the technology and availability of driverless cars may have a major impact on traffic safety and reducing the burden of PIs. 

## 6. Conclusions

In this study, we showed that PIs continue to be a burden for all SDI categories and corresponding countries. SDI categories show variation between incidence and mortality rates. Charters’ systematic review reveals the disparity in incidence and mortality between regions and countries and even between cities in the same country [25]. Although incidence and mortality are predicted to be reduced globally, according to our forecasting exercise, some specific SDI categories and their corresponding countries should be the primary focus because of their high mortality rates. Similar attention should be given to specific age groups. Accordingly, prevention efforts should be tailored for specific age groups, as noted in the discussion. Therefore, more research is needed to elaborate on specific problems and solutions in different SDI categories, regions, and countries. In summary, the incidence and mortality rates for road traffic injuries, specifically in pedestrians are still unacceptably high, and it is an issue that requires a multi-perspective approach in order to tackle the problem on a global scale.

## Figures and Tables

**Figure 1 ijerph-17-02135-f001:**
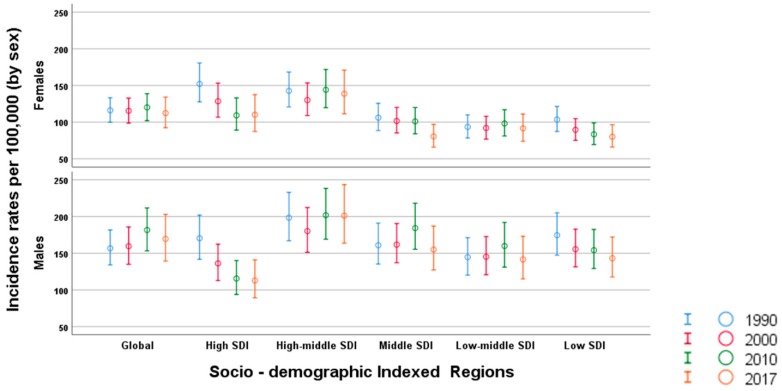
Incidence trends due to pedestrian road traffic injuries by sociodemographic indexed regions 1990–2017.

**Figure 2 ijerph-17-02135-f002:**
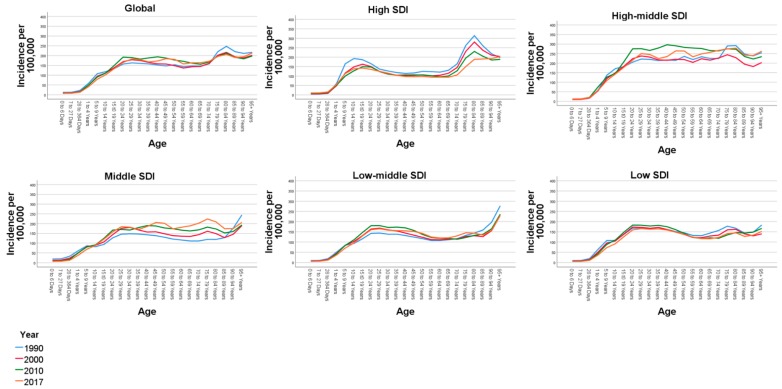
Trends for age-specific incidence rates for pedestrian road traffic injuries per 100,000 across sociodemographic indices 1990–2017. SDI: sociodemographic index.

**Figure 3 ijerph-17-02135-f003:**
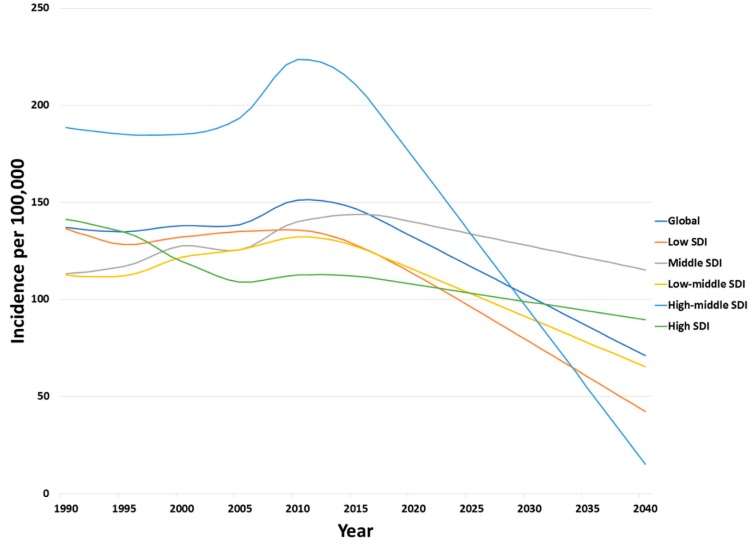
Forecasted trends for age-standardized incidence rates for pedestrian road traffic injuries per 100,000 population across sociodemographic indices 1990–2017.

**Figure 4 ijerph-17-02135-f004:**
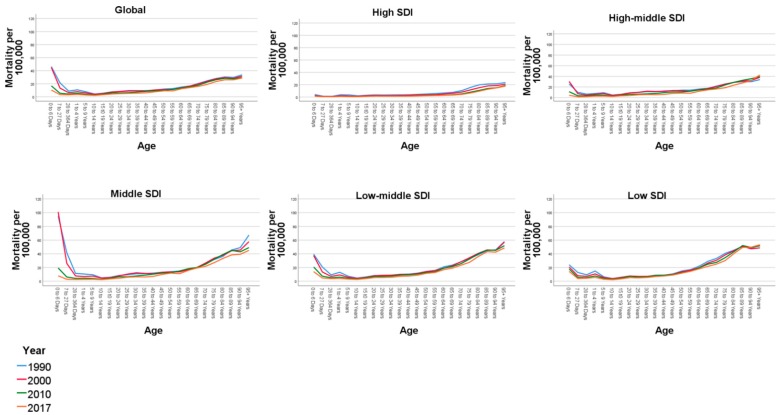
Trends for age-specific mortality rates for pedestrian road traffic injuries per 100,000 across sociodemographic indices 1990–2017.

**Figure 5 ijerph-17-02135-f005:**
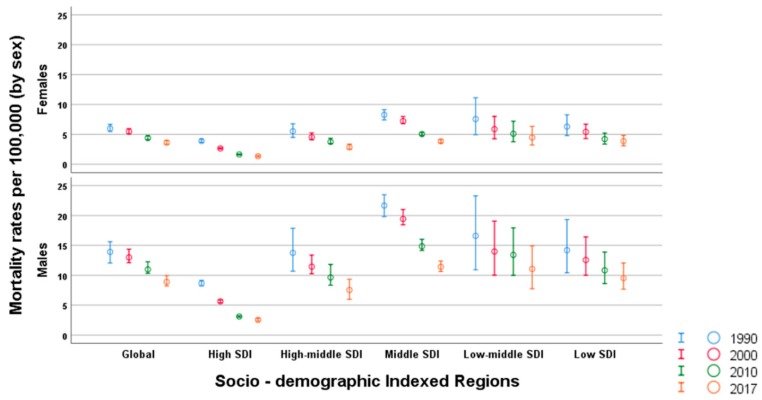
Mortality trends due to pedestrian road traffic injuries by sociodemographic indices 1990–2017.

**Figure 6 ijerph-17-02135-f006:**
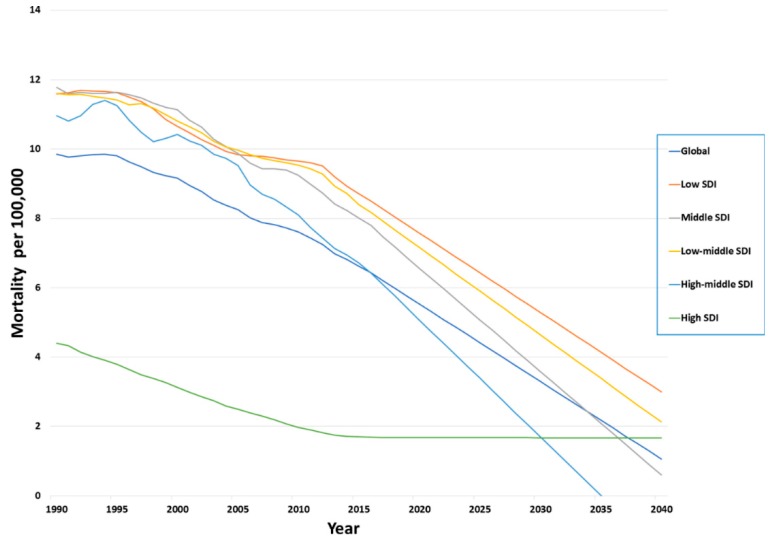
Forecasted trends for age-standardized mortality rates for pedestrian road traffic injuries per 100,000 across sociodemographic indices 1990–2017.

**Table 1 ijerph-17-02135-t001:** Incidence (A) and mortality rate (B) for pedestrian road traffic injuries (PIs) 1990–2017.

Category	A. PI Incidence (95% UI)			
	1990	2000	2010	2017
**Global**	136.4 (116.91–157.46)	137.49 (116.86–159.26)	150.85 (127.55–175.26)	140.92 (115.81–168.53)
**High SDI**	140 (118.69–163.47)	118.63 (100.3–138.74)	112.06 (93.04–133.03)	110.05 (88.33–135.33)
**High–middle SDI**	187.46 (159.72–216.89)	184.39 (156.05–213.86)	223.15 (188.96–259.89)	195.52 (160.55–234.45)
**Middle SDI**	113.08 (96.17–131.72)	127.27 (108.04–147.98)	140.04 (118.79–162.58)	143.19 (118–171.35)
**Low–middle SDI**	112.34 (95.31–131.22)	121.58 (103.23–142.46)	132.21 (110.98–155.84)	122.9 (100.75–147.7)
**Low SDI**	136.25 (116.2–158.91)	132.01 (112.5–153.71)	135.76 (114.39–160.5)	122.81 (101.48–147.67)
	**B.** **PI Mortality (95% UI)**			
**Global**	9.94 (8.75–11.14)	9.23 (8.59–10.16)	7.68 (7.21–8.5)	6.25 (5.77–6.94)
**High SDI**	4.47 (4.37–4.61)	3.16 (3.11–3.22)	1.99 (1.95–2.03)	1.69 (1.64–1.76)
**High–middle SDI**	11.1 (9.56–12.64)	10.54 (10–11.25)	8.2 (7.89–8.97)	6.14 (5.72–6.67)
**Middle SDI**	11.8 (10.35–13.23)	11.17 (10.47–12.22)	9.32 (8.84–10.06)	7.52 (6.98–8.18)
**Low–middle SDI**	11.57 (9.92–13.87)	10.82 (9.6–12.69)	9.59 (8.41–11.3)	7.99 (6.8–9.58)
**Low SDI**	11.48 (9.27–14.01)	10.6 (9.06–12.54)	9.64 (8.48–11.26)	8.29 (7.24–9.64)

PIs: pedestrian road traffic injuries, UI: uncertainty intervals, SDI: sociodemographic index. All figures are age-standardized rates per 100,000 population.

**Table 2 ijerph-17-02135-t002:** Incidence (A) and mortality trends (B) for pedestrian road traffic injuries (PIs) 1990–2017.

Category	A. Percentage Change in Incidence (95% UI)		
	1990 to 2017	2000 to 2017	2010 to 2017
**Global**	3.31 (−9.94 to 16.56)	2.49 (−9.99 to 14.97)	−6.58 (−14.25 to 1.09)
**High SDI**	−21.39 (−35.03 to −7.74)	−7.23 (−20.16 to 5.7)	−1.79 (−9.75 to 6.17)
**High–middle SDI**	4.29 (−9.81 to 18.39)	6.03 (−7.41 to 19.47)	−12.38 (−20.64 to −4.11)
**Middle SDI**	26.62 (8.87 to 44.36)	12.5 (−4.28 to 29.28)	2.24 (−8.25 to 12.73)
**Low–middle SDI**	9.4 (−6.93 to 25.73)	1.08 (−15.27 to 17.43)	−7.04 (−16.63 to 2.55)
**Low SDI**	−9.86 (−25.26 to 5.54)	−6.96 (−22.39 to 8.47)	−9.53 (−18.58 to −0.47)
	**B.** **Percentage Change in Mortality (95% UI)**		
**Global**	−37.12 (−45.19 to −29.04)	−32.28 (−40.31 to −24.24)	−18.61 (−23.32 to −13.89)
**High SDI**	−62.19 (−70.48 to −53.89)	−46.51 (−54.84 to −38.17)	−15.07 (−19.95 to −10.18)
**High–middle SDI**	−44.68 (−52.97 to −36.38)	−41.74 (−50.21 to −33.26)	−25.12 (−30.19 to −20.04)
**Middle SDI**	−36.27 (−43.23 to −29.3)	−32.67 (−42.32 to −23.01)	−19.31 (−25.92 to −12.69)
**Low–middle SDI**	−30.94 (−37.56 to −24.31)	−26.15 (−34.6 to −17.69)	−16.68 (−22.41 to −10.94)
**Low SDI**	−27.78 (−34.61 to −20.94)	−21.79 (−29.91 to −13.66)	−14 (−19.41 to −8.58)

PIs: pedestrian road traffic injuries, UI: uncertainty intervals, SDI: sociodemographic index. All figures are age-standardized rates per 100,000 population.

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
