# Peer review of "Global Incidence and Mortality Patterns of Pedestrian Road Traffic Injuries by Sociodemographic Index, with Forecasting: Findings from the Global Burden of Diseases, Injuries, and Risk Factors 2017 Study"

_ijerph, 2020, doi:10.3390/ijerph17062135_

Round 1

Reviewer 1 Report

This paper presents a study on the global incidence and mortality patterns of pedestrian road traffic injuries. The future trend of the pedestrian injuries is also forecasted.

The paper is premature and the reviewer cannot recommend publication of this paper.

The reviewer is concerned how the predictions in Fig.3 and 6 were made. Details are largely missing. Especially, in Fig.3, the “global trend” is in fact increasing if considering the data from 1990-2010.

The forecast model of the incidence trend is too simple – it is based on the analysis of existing database but has ignored many key factors. For example, how do authors consider the social/economic development for different regions? How do authors consider the development of regional transportation systems (road capacity, etc) with time? Ignoring these factors will make the study less convincing.

Reviewer 2 Report

The manuscript presents an interesting topic to address. Nevertheless, the authors are focused on data out of date. We are on 2020 and the last date is 2017. For this reason, I think they should include information about 2018 and 2019 to present a reasonable evaluation. This is the most pressing concern.

Moreover, the structure of the paper is very strange. The intro is not fluent and the background is not included and developed. The studies achieved are a little bit simple. I think the manuscript needs to be improved in a major revision.

Please take care of English and details. They provide the necessary quality of presentation.

I desire the best of lucks to authors in order to get an optimized releease of the manuscript.

Reviewer 3 Report

The manuscript is overall well-written. There are some minor typos or some small mistakes, but these are not enough to require   a rejection based on the writing and style alone.

If I can offer some suggestions, I noticed that sometimes the authors repeat themselves a little bit, in the sense that they used the same words over and over.  This happens both in some specific sentences, such as in lines 67 and 68 (“Age-standardized rates were used to allow comparisons of populations amongst different age groups across underlying 68 populations.” The word “population” is a little bit redundant), or in using the same structure to express a concept, such as in the case of the expression “due to”, which is used continuously throughout the manuscript (which is not a bad thing itself, but maybe you could utilize some synonyms, since it is not a technical term and there is plenty of other ways to express causality   - this could improve the readability of the text). Maybe you can take this as a piece of advice for any future article.

However, I would still like to compliment the authors for the overall result in what concerns the way they wrote and shaped their study.

Now, moving on to the contents:

In the abstract, please indicate which methods were employed, or which methodology was used for the projections. Additionally (even though it can seem obvious), I believe it is important to mention whether these predictions were used for everyone, taking into account the Global Burden of Disease from 1990 to 2017. Or, were the projections meant for a specific geographical area?

2.1. Data Source and Data Extraction: please state if the data on PIs from the GBD 2017 dataset is of public domain, or if you had to ask for permission in order to access it (please establish the ethical aspects).

Discussion: you mentioned “the 20 to 75 age group along with males 26 are highly prone to have a pedestrian injury in each SDI category”. Personally, this sentence sounds unfinished. It seems that the used method cannot differentiate between ages (basically the active life stage of any human being goes from the age of 20 to 75). In addition, this is not what appears in the graphics: I see an increase from the age of 70, both in mortality and incidence, and, besides, it is not linear, it has a down peak (or a stabilization) at the age of 80. Then it raises up again. It also seems like children and young people are quite affected. I suggest that you review this statement.

Round 2

Reviewer 1 Report

Authors should explain in Fig.3 why a decreasing trend is predicted for 2020-2040, given an increasing trend from 1990-2017, as this prediction is kind of counterintuitive. The reviewer acknowledges that a software has been used for the prediction but just concerns about the result.

The reviewer mentioned “how do authors consider the social/economic development for different regions? How do authors consider the development of regional transportation systems (road capacity, etc) with time?”. While it is, to some extent, ok that authors do not really take into these factors into account, authors should mention these factors in the “future work” part. The keyword is the “time-variation” of these factors, in the reviewer’s opinion.

Author Response

Authors should explain in Fig.3 why a decreasing trend is predicted for 2020-2040, given an increasing trend from 1990-2017, as this prediction is kind of counterintuitive. The reviewer acknowledges that a software has been used for the prediction but just concerns about the result.

Response: The incidence and mortality current data have been retrieved from 1990-2017 from the Global Burden of Disease database (GBD, 2017). We have used a total of 28 data sets points starting from 1990 to 2017, to compute forecasting trends till 2040. Although, PIs incidence increased by 3.31% in 2017 compared to 1990, the decline in incidence rates after 2010 is noteworthy. Starting from 1990, the global PIs incidence rates were similar until year 2005 (range, 136.4 to 138.5 per 100,000), then there was an increase until year 2011 (the highest being 151.3 per 100,000), and then again showed a decline until 2017 (140.9 per 100,000). Therefore, the analyses as shown in Figure 3 forecasted a decline in global PIs incidence rates across SDI regions until 2040.

We have incorporated these details now in the manuscript (Discussion: line 64-67).

The reviewer mentioned “how do authors consider the social/economic development for different regions? How do authors consider the development of regional transportation systems (road capacity, etc.) with time?”. While it is, to some extent, ok that authors do not really take into these factors into account, authors should mention these factors in the “future work” part. The keyword is the “time-variation” of these factors, in the reviewer’s opinion.

Response: We thank the reviewer for this suggestion. We have added now ‘Future work’ in our manuscript briefly discussing the role of time varying components of social/economic development together with the development of road and transportation systems.

Future Work

       Many parts of the world are undergoing economic and social development together with the development of road and transportation systems, including public transportation, which could have a major impact on the mortality and morbidity related to PIs. Moreover, national policies coupled with public health campaigns related to road and traffic safety are warranted to better educate both motorist and pedestrians. Future studies related to PIs should include time variations of these factors in their estimates. The advent of the technology and availability of driverless cars may have a major impact on traffic safety, and reducing the burden of PIs.

We have incorporated these details now in the manuscript (Future Work: line 131-138)

Reviewer 2 Report

The authors have tried to improve the quality of the manuscript successfully.

Author Response

Reviewer 2
The authors have tried to improve the quality of the manuscript successfully. (x) English language and style are fine/minor spell check required

Response: Thank you for your positive comments. We have made every effort to revise the whole manuscript thoroughly. Moreover the whole manuscript was revised by two independent native English speaking doctors whom we have acknowledged in the acknowledgement section.

Round 3

Reviewer 1 Report

The paper now can be accepted for publication.